# Systematic Review on Optical Diagnosis of Early Gastrointestinal Neoplasia

**DOI:** 10.3390/jcm10132794

**Published:** 2021-06-25

**Authors:** Andrej Wagner, Stephan Zandanell, Tobias Kiesslich, Daniel Neureiter, Eckhard Klieser, Josef Holzinger, Frieder Berr

**Affiliations:** 1Department of Internal Medicine I, University Clinics Salzburg, Paracelsus Medical University, Müllner Hauptstrasse 48, 5020 Salzburg, Austria; s.zandanell@salk.at (S.Z.); t.kiesslich@salk.at (T.K.); frieder.berr@pmu.ac.at (F.B.); 2Laboratory for Tumour Biology and Experimental Therapies (TREAT), Center for Physiology, Pathophysiology and Biophysics—Salzburg and Nuremberg, Institute for Physiology and Pathophysiology—Salzburg, Paracelsus Medical University, 5020 Salzburg, Austria; 3Institute of Pathology, University Clinics Salzburg, Paracelsus Medical University, 5020 Salzburg, Austria; d.neureiter@salk.at (D.N.); e.klieser@salk.at (E.K.); 4Cancer Cluster Salzburg, 5020 Salzburg, Austria; 5Department of Surgery, University Clinics Salzburg, Paracelsus Medical University, 5020 Salzburg, Austria; J.Holzinger@salk.at

**Keywords:** endoscopy, neoplasia, magnification endoscopy, chromoendoscopy, invasion depth

## Abstract

Background: Meticulous endoscopic characterization of gastrointestinal neoplasias (GN) is crucial to the clinical outcome. Hereby the indication and type of resection (endoscopically, en-bloc or piece-meal, or surgical resection) are determined. By means of established image-enhanced (IEE) and magnification endoscopy (ME) GN can be characterized in terms of malignancy and invasion depth. In this context, the statistical evidence and accuracy of these diagnostic procedures should be elucidated. Here, we present a systematic review of the literature. Results: 21 Studies could be found which met the inclusion criteria. In clinical prospective trials and meta-analyses, the diagnostic accuracy of >90% for characterization of malignant neoplasms could be documented, if ME with IEE was used in squamous cell esophageal cancer, stomach, or colonic GN. Conclusions: Currently, by means of optical diagnosis, today’s gastrointestinal endoscopy is capable of determining the histological subtype, exact lateral spread, and depth of invasion of a lesion. The prerequisites for this are an exact knowledge of the anatomical structures, the endoscopic classifications based on them, and a systematic learning process, which can be supported by training courses. More prospective clinical studies are required, especially in the field of Barrett’s esophagus and duodenal neoplasia.

## 1. Introduction

Gastrointestinal carcinomas have the highest incidence (19.2% of all new cancer cases) and mortality worldwide (23% of annual worldwide cancer mortality, 1.81 million [1]). As a result of an increase in endoscopic (screening) examinations and the continuous improvement of the optical quality of endoscopic equipment, T1 carcinomas in the gastrointestinal tract (GIT) are increasingly detected and diagnosed. However, this group is very heterogeneous with regard to the risk of lymph node metastasis (LM): e.g., 7–17% of colonic T1 carcinomas already have LM [2], which significantly depends on the depth of invasion. Therefore, correct endoscopic assessment of the depth of invasion of these lesions has a great prognostic and therapeutic importance. It allows for the decision for endoscopic en-bloc resection or primary surgical oncologic resection with lymph node dissection, respectively. In this context, the term “early carcinoma” implicates locally curatively resectable lesions with a very low risk of LM. Biopsy, although often an indispensable tool in this decision-making, has the disadvantage of a sampling error, yielding discordant results in up to 40% in the colon [3]. In a recently published study [4], only 39% of all T1 carcinomas in the colon were correctly detected endoscopically. In the remaining cases, inadequate piece-meal ablation led to (avoidable) adjuvant surgical therapy in 41% of cases. This number was significantly lower (11%) in the group of correctly characterized T1 carcinomas.

High definition (HD) endoscopy, which has been available since 2005, has a 35-fold magnification. However, with aids such as attachment caps, underwater magnification and digital zoom, magnifications of up to 75-fold can be achieved with conventional HD endoscopes. Magnification endoscopy (ME) in HD endoscopes with optical extensions, can achieve a 60-fold to 150-fold magnification. Complemented by contrast-enhanced endoscopy (image-enhanced endoscopy, IEE) based on staining (true chromoendoscopy, CE) or device-based, virtual chromoendoscopy (e.g., as narrow-band imaging, NBI or blue light imaging, BLI), the accuracy of endoscopic characterization of lesions in the GIT could be improved dramatically. Device-based IEE, such as NBI, enables improved contrasting of vascular structures by emission of blue and green light of a defined wavelength (in the absorption range of hemoglobin). Practically, these technical enhancements visualize anatomical microstructures in the endoscopic image. Based on changes in the microsurface structure (SP) and the microvascular pattern (capillary structures, VP), numerous classifications of pathological changes have been published in the various areas of the GIT in recent years, the most important of which will be presented and evaluated in terms of evidence and practical application in this review.

## 2. Examination Procedure—Detection and Characterization

A high quality of the endoscopic examination (e.g., good bowel preparation, mucolytic/surface preparation of the mucosa, meticulous examination technique, standardized examination protocols) is the prerequisite for reliable detection and characterization of suspicious lesions [5]. First, at the macro-level of screening, conventional white-light endoscopy (WLE) is used to assess the color and structural changes and to determine the morphology of a lesion. Secondly, in the phase of the micro examination, SP and VP are assessed. Already in the first step, on the basis of morphology and certain additional features (e.g., sharp delineation, inhomogeneity of color distribution), reliable conclusions about the endoscopic resectability of the lesion can often be made, not requiring ME (NICE classification in the colon and rectum [6,7], GUP system in the stomach [8]). The Paris-Japanese classification and, in addition, the LST classification in the colon have become widely accepted throughout the GI tract [9,10]. Thus, more than 50% of LST (lateral spreading tumors) of the non-granular type in the colon and up to 80% of all flat (0-II) lesions in the stomach are associated with malignant histology [11,12]. To further characterize gastrointestinal neoplasms (dignity, lateral extent, and depth of invasion), VP and (ideally with CE) SP are always assessed in ME (Figure 1).

## 3. Materials and Methods

To answer the primary research question, “With what statistical accuracy can GN be characterized visually-endoscopically?”, the present review is based on a systematic literature search in the databases PubMed, Cochrane Library, and the International Standard Randomised Controlled Trial Number (ISRCTN) registry using the search terms “early cancer detection”, “diagnosis”[Mesh] OR “diagnostic imaging”[Mesh] OR “pathology”[Mesh] and “endoscopy, gastrointestinal” for relevant publications in the period from 1 January 2000 to 1 May 2021. Only prospective studies and meta-analyses were included providing information on the statistical accuracy of optical characterization of GN with a histopathologic evaluation of the resected lesions as the gold standard in at least 20 cases. The bibliographies of the cited studies were additionally searched for further relevant studies. In addition, individual older publications, textbooks, and the clinical experience of the authors were considered and indicated as such where applicable. Included studies are listed in Table 1. The search strategy is depicted in Appendix A. This review conforms to the PRISMA guidelines (Figure 2, [13]).

Developments in the field of endomicroscopy and artificial intelligence are not discussed, as these promising methods are currently reserved for individual, specialized centers or because there are only a few clinical studies available. For Barrett’s esophagus and duodenal neoplasms, the limited data available do not yet allow a statistically reliable assessment of the depth of invasion [35]. Endoscopic ultrasound performed during diagnostic endoscopy allows for T-staging of early cancer and on subepithelial lesions arising in intramural echo structures with high accuracy, especially, when high-resolution probes are used [36]. In squamous cell lined esophagus, stomach and rectum both techniques it is recommended to combine both techniques in selected cases. However, this issue is beyond the scope of this review. Also, (pre-)malignant changes in the context of inflammatory bowel disease and characterization of diminutive colorectal polyps are not discussed here.

## 4. Results

### 4.1. Esophagus

In the squamous esophagus, sessile/polypoid (0-Is/p) or ulcerative neoplasms (0-III) have the highest risk of deep submucosal invasion (SMI, >80%). However, even shallow lesions (0-II) already have a risk of SMI of 15–53% [37]. Similarly, in Barrett’s esophagus, up to 45% of 0-IIa lesions already represent T1b carcinomas [38]. Therefore, a closer characterization at the micro-level by combining ME and IEE is required. Only two studies in the context of squamous cell carcinoma (SCC) met the inclusion criteria (Table 1). The recently published JES classification of the Japanese Esophageal Society, which is a further development of the IPCL classification [39], showed a statistical accuracy of 91% regarding the depth of invasion in a prospective study [15]. This was also confirmed in a recent meta-analysis [14]. This classification is based on the pattern of microvessels (VP) and their degree of dilatation. Four VP types are defined, which allows for a subtle assessment of the degree of submucosal invasion of SCC in the esophagus. Only intramucosal SCC is appropriate for endoscopic resection (JES type B1). Lesions invading the muscularis mucosae or the upper layer of the submucosa (JES type B2) are a relative contraindication for endoscopic resection.

Also in Barrett’s esophagus, ME-NBI (for VP) and acetic acid-based ME-CE (for SP) can identify areas of high-grade intraepithelial neoplasia or early carcinoma with more than 90% sensitivity and specificity, and low interobserver variability [16,17,40,41]. Only two studies in the context of Barrett’s carcinoma met the inclusion criteria (BING classification validation study and simplified NBI classification, Table 1). The most recent of the existing classifications of Barrett’s esophagus is the JES-BE classification, which applies the clinically well-studied criteria in gastric carcinoma to the changes in Barrett’s carcinoma [42]. In brief, similarly to the BING classification, mucosal and vascular patterns are classified separately (regular or irregular). In the JES-BE classification, visibility and morphologic features (e.g., pit or network) are specified additionally. This classification reflects the natural highly variable appearance of Barrett’s lesions. Of note, this classification has not yet been validated in clinical prospective studies. However, quadrant biopsies according to the Seattle protocol cannot be dispensed, as about half of early malignant neoplasms are detected by “protocol biopsies” [42,43]. Assessment of depth of invasion is challenging in Barrett’s (early) carcinoma—in practice, a combined view of morphology and SP/VP is recommended. Deep SMI is very likely in all sessile (0-Is), elevated (0-IIa), sunken (0-IIc), or ulcerated (0-III) lesions in combination with a highly irregular VP (i.e., high variability of vessel diameters, low density) and/or amorphous or very irregular SP [35].

### 4.2. Stomach

In order to achieve the highest detection rates and accuracy of characterization, a combination of meticulous WLE followed by ME-NBI is currently recommended [44]. Already with WLE, a high quality of detection of suspicious lesions can be achieved after good preparation, taking into account the medical history (risk factors such as helicobacter pylori infection). Here, a standardized systematic screening protocol is recommended [45]. Localization, color, demarcability to the surrounding mucosa, and morphology should be analyzed for each suspicious lesion. However, beyond WLE, the ME-NBI technique which allows analysis of the specific anatomical structures (e.g., subepithelial capillaries and marginal crypt epithelium), especially in small gastritis-like lesions, is crucial for characterization. Eight studies and meta-analyses met the inclusion criteria (Table 1), showing a very high diagnostic accuracy (>95%) of ME-NBI in characterization in the context of early gastric cancer. Figure 3 and Figure 4 illustrate this diagnostic process and the correlations with histologic criteria (differentiated early carcinoma in the antrum). The “vessel plus surface” classification enables differential diagnosis of differentiated early carcinoma (sharp demarcation in combination with irregular SP and/or VP) with a sensitivity and specificity of 95%, which was confirmed in a recently published meta-analysis [20]. In this classification, three categories exist for SP and VP, respectively: regular, irregular, and absent. The presence of an irregular VP with a demarcation line, or the presence of an irregular SP with a demarcation line is diagnostic for early gastric cancer. In this context, the “white zone” is equivalent to the gastric SP, anatomically correlated to the marginal crypt epithelium. In Figure 3 and Figure 4, all features of the vessel plus surface classification are shown for a neoplastic lesion and normal mucosa with corresponding histopathological sections, respectively. Weaknesses exist in the diagnosis of undifferentiated gastric carcinomas (G3, G4), which may appear as pale, sunken lesions on WLE, and may often have poorly defined delineation and, in some circumstances, an inconspicuous SP (if diffusely subepithelial) on ME-NBI. The depth of invasion of differentiated carcinomas is reliably assessed by macroscopic morphology (Paris type, fold criteria), VP, and SP. Thus, a raised or sunken/ulcerated lesion with loss of SP has a >80% risk of massive SMI (≥SM2, [46]).

### 4.3. Colon

Nine prospective studies and meta-analyses met the inclusion criteria (Table 1), showing a very high diagnostic accuracy (>90%) of ME-NBI in characterization in the context of early colorectal cancer. To achieve >90% statistical confidence in the characterization of colorectal neoplasms it is essential to evaluate colorectal neoplasms according to VP (Sano or JNET, in ME-NBI) and SP (pit pattern according to Kudo, in ME-CE [32,46,47]). In Figure 5, two examples of endoscopic diagnosis (WLE, NBI, and CE) of colorectal lesions are demonstrated. Both lesions show a centrally sunken surface when examined by WLE alone (IIa + IIc). ME-NBI shows an irregular dense or highly irregular rarefied vascular pattern (VP) in the central area, respectively. Supplementary CE allows for the assessment of the VP (irregular, high density, or amorphous surface). JNET classification is based on this analysis algorithm to distinguish hyperplastic, adenomatous, superficial malignant, and deeply invasive lesions [48]. In brief, for each lesion VP and SP are evaluated separately. Variable caliber or irregular distribution of VP and/or irregular or obscure SP are diagnostic for high-grade intramucosal neoplasia or shallow submucosal cancer (JNET Type 2B). Loose vessel areas or interruption of thick vessels and/or amorphous SP are diagnostic for deep submucosal invasive cancer (JNET Type 3), not suitable for endoscopic resection. Morphological criteria such as nodular formations on polyps, amorphous color changes, bleeding tendency, and central ulceration with lack of surface structure, as well as fixed deformities after desufflation have high importance in terms of estimation of deep SMI (>1000 µm) and endoscopic non-resectability. In this regard, analysis by ME-NBI supported by CE (crystal violet, indigo carmine) could show sensitivities of 84–90% and a pooled specificity of 98% in a recent meta-analysis (independent of the classification system used). This is significantly superior to a mere WLE diagnosis [27,33]. The NBI international colorectal endoscopic classification (NICE), which utilizes NBI *without* ME, allowed for a specificity of >96% in identifying invasive, i.e., endoscopically unresectable T1 colorectal tumors [31].

## 5. Discussion

Early carcinomas are increasingly detected in the gastrointestinal tract, with T1 carcinomas already showing lymph node metastases in up to 17%. Here, the depth of invasion is a major risk factor. The accuracy in the characterization of colorectal neoplasia using solely WLE in standard definition (SD) is 59% [49]. Endoscopic optical diagnostics with the aid of magnification endoscopy (ME), true (CE) and so-called virtual chromoendoscopy (e.g., NBI or BLI) can determine histology and depth of invasion with very high accuracy (>90%) and thus overcome the so-called sampling error or discordant results of a biopsy. Based on this assessment and supplemented by other imaging modalities (EUS, radiological diagnostics), an essential decision in the therapy of early carcinomas can thus be made: curative endoscopic resection, or surgical resection with lymph node dissection, respectively. Of note, also detection rates of adenomas in surveillance colonoscopy can be massively improved by advanced imaging techniques [50].

However, although the means of endoscopic visual diagnosis at the HD level allow very accurate characterization and are now widely available, the rate of mischaracterized early carcinomas in the GIT is alarmingly high. There are some weaknesses in the endoscopic diagnosis of colorectal lesions, and the characterization of colorectal lesions is of suboptimal quality [4]. In the prospective multicenter study by Vleugels, endoscopists correctly diagnosed T1 colorectal cancers in only 39% of 92 cases, which led to significant differences in treatment outcomes. Another prospective clinical trial comparing diagnostic accuracy in colorectal lesions > 10 mm using the NICE classification showed a sensitivity of 58% and a specificity of 96% to determine invasive colorectal tumors. The weakness of the NICE classification is the lack of a subcategory, identifying lesions with high-grade dysplasia or superficial submucosal invasive carcinomas, suitable for en-bloc endoscopic submucosal dissection. This may bear the risk of piece meal resection of early colorectal carcinomas. Of note, this undertreatment of malignant lesions by piece-meal resection led to the differences in the clinical outcome in the study by Vleugels [4]. The JNET classification, which utilizes ME-NBI, allows identifying such lesions (JNET category 2 B). However, the accuracy of identifying Type 2B-lesions was lower (75%) than those of other JNET categories [30]. Similarly, there are diagnostic limitations in the optical diagnosis of undifferentiated gastric adenocarcinoma and a missing demarcation line in cases after eradication of helicobacter pylori [46,51,52].

In experienced hands, endoscopic optical characterization of early gastrointestinal neoplasia has shown to be feasible and very helpful in clinical practice. However, the performance varies substantially among endoscopists, depending on their experience and a learning process that can be supported by seminars, e-learning, and new bioinformatical methods [53]. A recent initiative by the European Society of Gastrointestinal Endoscopy (ESGE) focuses on the criteria for optical diagnosis proficiency and training methods. Their position statement [54] defines the prerequisites for competency, proficiency, and competence thresholds in optical diagnosis. However, in a former guideline, ESGE suggests that routine use of advanced imaging for detection and differentiation of colorectal neoplasia in average-risk patients has to be balanced against costs and practical considerations [55].

## 6. Conclusions

Currently, by the means of optical diagnosis, today’s gastrointestinal endoscopy is capable to determine the histological subtype, exact lateral spread, and depth of invasion of a lesion, especially of the esophagus, stomach, and colon. Magnification and (virtual) chromoendoscopy, which have been increasingly available in the past years, have a statistical accuracy of more than 90%, which is essential to select the optimal oncological therapy strategy in early gastrointestinal neoplasia. The prerequisites for this are an exact knowledge of the anatomical structures, the endoscopic classifications based on them, and a systematic learning process, supported by training courses. Nevertheless, there is only limited data on the endoscopic characterization of duodenal neoplasms and in the field of Barrett’s esophagus in the literature. More prospective clinical studies are required.

## Figures and Tables

**Figure 1 jcm-10-02794-f001:**
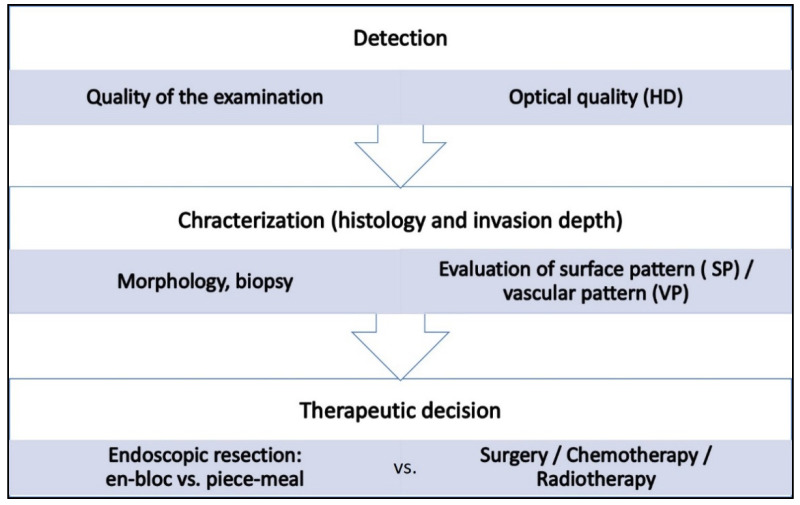
The steps of the diagnostic process in endoscopy, the according key issues and their impact on therapeutic decisions.

**Figure 2 jcm-10-02794-f002:**
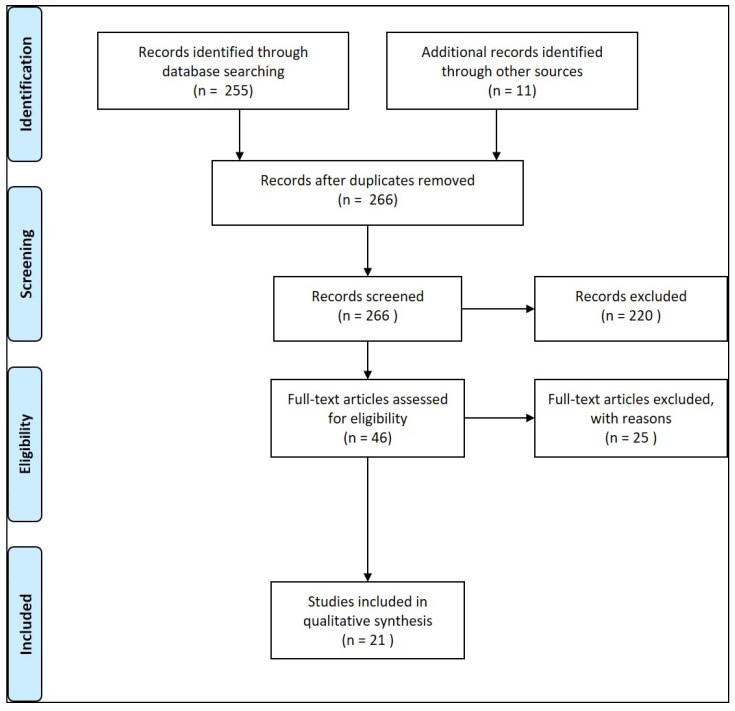
Flow chart of the literature search.

**Figure 3 jcm-10-02794-f003:**
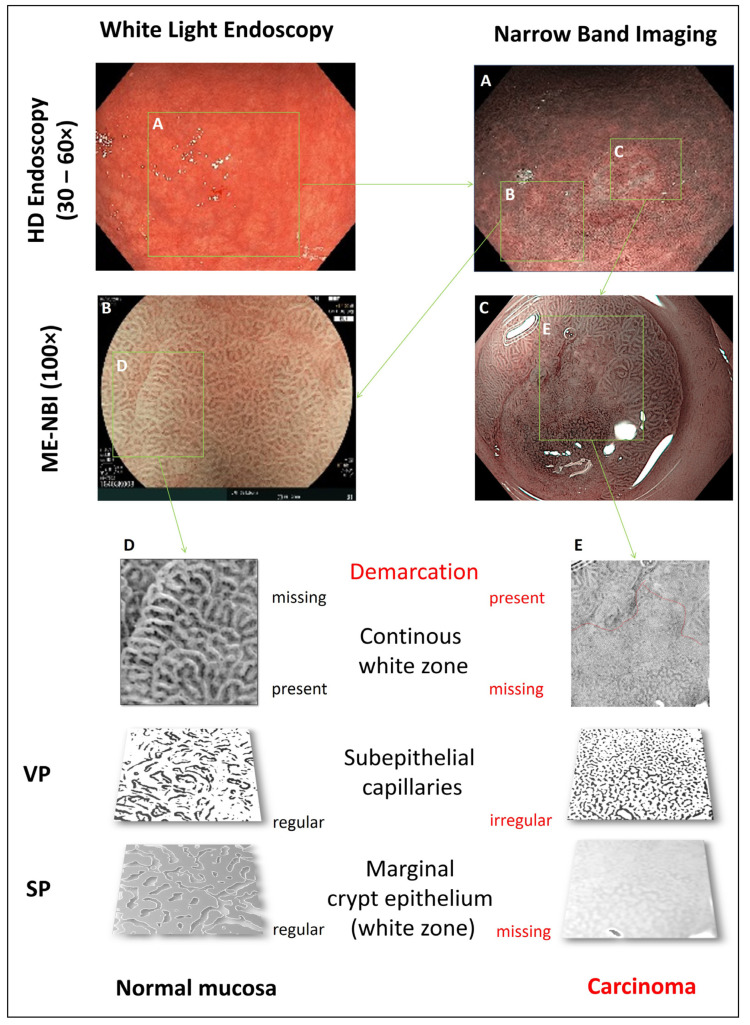
Optical endoscopic characterization of a gastric lesion (antrum). (**A**): The lesion is merely demarcated in white light HD endoscopy. Magnification endoscopy (ME) with narrow band imaging (NBI) allows for characterization of normal vs. cancerous areas in the antrum (**B**,**C**). Determining factors are demarcation and presence of a white zone. Vascular pattern (VP) is determined by the pattern of subepithelial capillaries, and surface pattern (SP) by the marginal crypt epithelium (i.e., the “white zone”), respectively (**D**,**E**).

**Figure 4 jcm-10-02794-f004:**
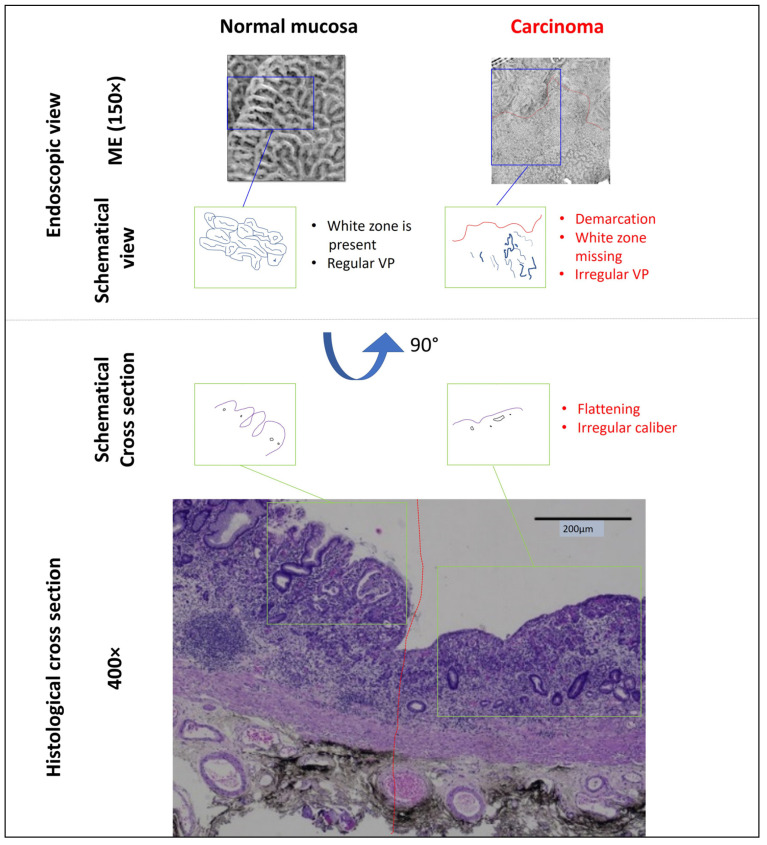
Comparison of magnification endoscopy view (ME) and histological cross section of normal mucosa and an early well differentiated adenocarcinoma in the antrum (compare Figure 2). The components of the ME picture, vascular pattern (VP) and surface pattern (SP), represent marginal crypt epithelium (i.e., the “white zone”) and subepithelial capillaries, respectively. SP and white zone are missing in the malignant area, VP is irregular. This corresponds to a flattening of the epithelium and irregular caliber of the capillary vessels in the histological specimen (obtained after endoscopic submucosal dissection of the lesion, which is characterized in Figure 2).

**Figure 5 jcm-10-02794-f005:**
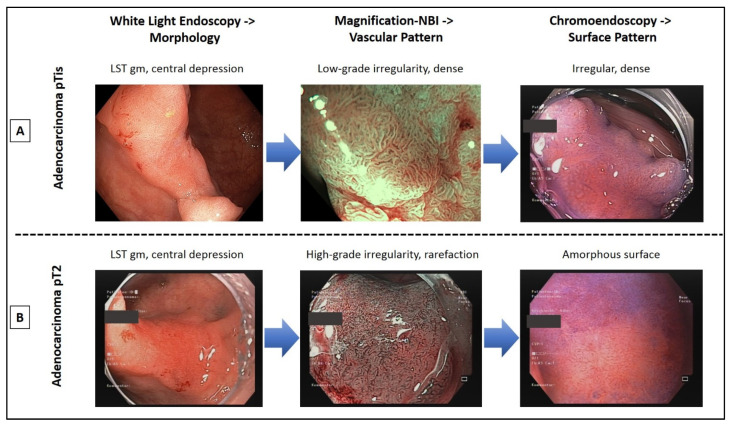
Endoscopic optical characterization of two large colonic lateral spreading tumors of the granular mixed type (LST gm). Magnification endoscopy with narrow band imaging (NBI) and chromoendoscopy with crystal violet allow for differentiation between early adenocarcinoma, which can be curatively resected by endoscopic submucosal dissection (**A**) and invasive carcinoma, requiring oncologic surgery (**B**).

**Table 1 jcm-10-02794-t001:** Studies obtained by the systematic literature search on optical diagnosis of gastrointestinal neoplasia.

Imaging Technique	Study Type	*n* st.	*n* Lesions	Accuracy (%)	Sensitivity (%)	Specificy (%)	First Author	Year	Ref.
**Esophagus, squamous cell carcinoma, T1a vs. T1b**									
ME-NBI vs. WLE, JES	Meta-A.	10	1033		90	90	Yu	2018	[14]
ME-NBI, JES	Prosp.		200	91			Oyama	2017	[15]
**Barrett’s esophagus, high grade dysplasia/early adenocarcinoma**									
NBI, BING	Prosp.		120	85	91	93	Sharma	2016	[16]
ME-NBI, simplif.	Prosp.		248	95			Kato	2017	[17]
**Stomach, early gastric cancer**									
NBI vs. WLE, simplif.	Prosp.		238	94	92		Pimentel-Nunes	2012	[18]
ME-NBI vs. WLE, SD	Meta-A.	10	2151		87 ^a^	93 vs. 65	Lv	2015	[19]
ME-NBI	Meta-A.	9	5398		88	96	Zhou	2018	[20]
ME-NBI vs. WLE, SD	Rand., Prosp.		40	97	95	97	Ezoe	2011	[21]
ME-NBI vs. WLE, SD	Meta-A.	10	2153	96	83	96	Zhang	2015	[22]
ME-NBI vs. WLE, SD	Meta-A.	14	2171		86	96	Hu	2015	[23]
ME-NBI, VS	Rand., Prosp.		1097	98	86	99	Yao	2014	[24]
ME-NBI, VS	Syst. Rev.	66		95	PPV 79	NPV 99	Muto	2016	[25]
**Colorectum, early CRC**									
ME(-NBI) vs. WLE ^b^	Meta-A.	13			80 ^b^		Parikh	2016	[26]
ME-NBI vs. ME-CE ^e^	Meta-A.	17		97	84 ^c^	97 ^c^	Zhang	2017	[27]
ME ^d^	Meta-A.	20	5111	94	89	96	Li	2014	[28]
ME-NBI, JNET	Prosp.		40	90			Minoda	2019	[29]
ME-NBI, JNET	Prosp.		2933	97	55	100	Sumimoto	2017	[30]
NBI, NICE	Prosp.		2123		58	96	Puig	2018	[31]
SP and/or VP ^d^	Meta-A.	36	9607	99 ^f^			Guo	2018	[32]
ME-NBI, -CE vs. WLE ^e^	Meta-A.	33	31,568		90		Backes	2017	[33]
NBI and WLE ^e^	Prosp.		343		79		Backes	2019	[34]

Abbreviations: st.: studies, Ref.: reference, ME: magnification endoscopy, NBI: narrow band imaging, WLE: white light endoscopy, JES: classification of the Japan esophageal society, Meta-A.: Meta-Analysis, prosp.: prospective, BING: Barrett’s International NBI Group, simplif.: simplified NBI classification, SD: standard definition, Rand.: randomized, prosp.: prospective, VS: vessel plus surface-classification, Syst. Rev.: systematic review, PPV: positive predictive value, NPV: negative predictive value, CRC: colorectal cancer, CE: chromoendoscopy, JNET: Japan NBI expert team classification, NICE: narrow-band imaging international colorectal endoscopic classification, SP: surface pattern, VP: vascular pattern. ^a^ vs. 61%, WLE; ^b^ ME-NBI, 60% f. NBI, sessile serrated adenoma vs. Adenoma vs. hyperplastic polyp; ^c^ ME-CE, invasive CRC; ^d^ neoplastic lesion; ^e^ invasive CRC; ^f^ SP and VP combined.

## Data Availability

Exclude this statement.

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
