# Peer review of "Systematic Review on Optical Diagnosis of Early Gastrointestinal Neoplasia"

_jcm, 2021, doi:10.3390/jcm10132794_

Round 1

Reviewer 1 Report

The review is comprehensive and concisely communicates the essential contents of modern endoscopic diagnostic procedures. However, it would be helpful if the authors would again briefly describe the essential diagnostic criteria that are relevant for the individual classifications presented. We assume that not all readers are familiar with the details of the classifications.

Author Response

Dear Reviewer,

Revision Manuscript jcm-1262878             

Thank you very much for handling our manuscript “Systematic Review on Optical Diagnosis of Early Gastrointestinal Neoplasia” by Andrej Wagner et al.

We fully agree with the comments and have revised our manuscript incorporating all suggested changes – please see below for our point-by-point replies.

We think the comments have significantly improved the quality of the manuscript and hope that the manuscript might now be suitable for publication in Journal of Clinical Medicine.

REVIEWER 1:

The review is comprehensive and concisely communicates the essential contents of modern endoscopic diagnostic procedures. However, it would be helpful if the authors would again briefly describe the essential diagnostic criteria that are relevant for the individual classifications presented. We assume that not all readers are familiar with the details of the classifications.

Thank you for the important comment. Essential diagnostic criteria have now been described in more detail and are now provided in the manuscript (“results”), lines 152-157, 165-168, 194-200, and 234-239.

Extensive editing of English language and style required.

The manuscript has been checked by a native English-speaking colleague. Accordingly, we have modified and amended the manuscript.

Please see the attachment for the revised version of the manuscript.

Reviewer 2 Report

The authors present a systematic review of the literature in order to characterize gastrointestinal neoplasia in terms of malignancy and invasion depth  by means of established image-enhanced (IEE) and magnification endoscopy (ME). 21 studies were found which met the inclusion criteria. In clinical prospective trials and meta-analyses, a diagnostic accuracy of > 90% for characterization of malignant neoplasms is  documented, if ME with IEE was used in squamous cell esophageal cancer, stomach or colonic neoplasms.  They concluded that by means of optical diagnosis, today's gastrointestinal endoscopy is capable to determine the histological subtype, exact lateral spread, and depth of invasion of the lesions. More prospective clinical studies are required in the fields of Barrett's oesophagus and duodenal neoplasia.

The topic of the study is interesting and potentially very important for the appropriate treatment of early gastrointestinal neoplasms.

I have a major concern about the reproducibility of these methods and conclusions together their feasibility in clinical practice. The role of echoendoscopy should be redefined?

Minor points:

  • Duodenal neoplasms are not included in this review but the Authors suggest more prospective clinical studies.
  • A flow-chart of literature's search according to PRISMA guidelines should be included in the text.

Author Response

Dear Reviewer,

Revision Manuscript jcm-1262878             

Thank you very much for handling our manuscript “Systematic Review on Optical Diagnosis of Early Gastrointestinal Neoplasia” by Andrej Wagner et al.

We fully agree with the comments and have revised our manuscript incorporating all suggested changes – please see below for our point-by-point replies.

We think the comments have significantly improved the quality of the manuscript and hope that the manuscript might now be suitable for publication in Journal of Clinical Medicine.

REVIEWER 2:

The authors present a systematic review of the literature in order to characterize gastrointestinal neoplasia in terms of malignancy and invasion depth by means of established image-enhanced (IEE) and magnification endoscopy (ME). 21 studies were found which met the inclusion criteria. In clinical prospective trials and meta-analyses, a diagnostic accuracy of > 90% for characterization of malignant neoplasms is documented, if ME with IEE was used in squamous cell esophageal cancer, stomach or colonic neoplasms. They concluded that by means of optical diagnosis, today's gastrointestinal endoscopy is capable to determine the histological subtype, exact lateral spread, and depth of invasion of the lesions. More prospective clinical studies are required in the fields of Barrett's oesophagus and duodenal neoplasia.

The topic of the study is interesting and potentially very important for the appropriate treatment of early gastrointestinal neoplasms.

I have a major concern about the reproducibility of these methods and conclusions together their feasibility in clinical practice. The role of echoendoscopy should be redefined?

Thank you for this important remark. We fully agree, that clinical feasibility of optical diagnosis in endoscopy is a challenging issue. The rate of mischaracterized early carcinomas in the GIT is alarmingly high 1. It was our aim to clarify this point in this review, and therefore we aimed to evaluate evidence and usefulness of high definition (HD) endoscopy with magnification complemented by contrast-enhanced endoscopy (IEE). These technical developments are commercially available for more than ten years and their use is emerging. In a recent unpublished small survey among 40 Austrian resident endoscopists, who joined our seminar, 60% routinely use HD endoscopy, and 40% use magnification / NBI. However, the aspect of training and experience is still a critical issue, and it is addressed by the recent ESGE initiative in optical diagnosis 2.

We fully agree that the issue of practical application of optical diagnosis has to be described in a more precise manner: We have discussed these aspects more critical in the revised version of the manuscript (“discussion”, lines 288-291).

Echoendoscopy (EUS) is an indispensable tool in endoscopic staging of gastrointestinal neoplasia 3. We did not address this topic, because it is beyond the scope of the review. Unfortunately, we missed to clarify this point in the manuscript. As a consequence, we added a passage to the “materials” section (lines 133-138).

Minor points:

Duodenal neoplasms are not included in this review but the Authors suggest more prospective clinical studies.

Thank you for the comment. There is only limited data on endoscopic characterization of duodenal neoplasms in the literature. Available studies in this field are retrospective with a limited number of cases. Therefore, we suggested to perform more clinical studies in this field with the goal to establish a diagnostic system. To clarify this point, we added a passage in the “conclusions” section (lines 302-310)

A flow-chart of literature's search according to PRISMA guidelines should be included in the text.

Thank you for the comment. We included the flow chart of the search in the manuscript (figure 2).

Please see the attachment for a revised version of the manuscript

  1. Vleugels, J.L.A.; Koens, L.; Dijkgraaf, M.G.W.; Houwen, B.; Hazewinkel, Y.; Fockens, P.; Dekker, E.; group, D.s. Suboptimal endoscopic cancer recognition in colorectal lesions in a national bowel screening programme. Gut 2020, 69, 977-980.
  2. Dekker, E.; Houwen, B.; Puig, I.; Bustamante-Balen, M.; Coron, E.; Dobru, D.E.; Kuvaev, R.; Neumann, H.; Johnson, G.; Pimentel-Nunes, P., et al. Curriculum for optical diagnosis training in europe: European society of gastrointestinal endoscopy (esge) position statement. Endoscopy 2020, 52, 899-923.
  3. Seifert, H.; Kikuchib, D.; Yahagi, N. Colorectum: High-Resolution Endoscopic Ultrasound: Clinical T-Staging of Superficial and Subepithelial Gastrointestinal Neoplasias. In Atlas of early neoplasias of the gastrointestinal tract: Endoscopic diagnosis and therapeutic decisions, 2nd ed.; Berr, F.; Oyama, T.; Ponchon, T.; Yahagi, N., Eds. Springer: 2019, p.79.

Round 2

Reviewer 2 Report

I think that the Authors adequately replie to the reviewers' comments and the manuscript has been improved in this revised form